# Three Decades of Land Cover Change in East Africa

Eric L. Bullock [1,*], Sean P. Healey [1], Zhiqiang Yang [1], Phoebe Oduor [2], Noel Gorelick [3], Steve Omondi [2], Edward Ouko [2] and Warren B. Cohen [4]

1   US Forest Service, Rocky Mountain Research Station, Ogden, UT 84401, USA; sean.healey@usda.gov (S.P.H.); zhiqiang.yang@usda.gov (Z.Y.)

2   Regional Centre for Mapping of Resources for Development, Nairobi 00618, Kenya; poduor@rcmrd.org (P.O.); sotieno@rcmrd.org (S.O.); eouko@rcmrd.org (E.O.)

3   Google Switzerland, 8002 Zurich, Switzerland; gorelick@google.com

4   USDA Forest Service, Pacific Northwest Research Station, 3200 SW Jefferson Way, Corvallis, OR 97331, USA; hammerfiddle@gmail.com

*   Correspondence: eric.bullock@usda.gov; Tel.: +1-603-801-0135

**Abstract:** Population growth rates in Sub-Saharan East Africa are among the highest in the world, creating increasing pressure for land cover conversion. To date, however, there has been no comprehensive assessment of regional land cover change, and most long-term trends have not yet been quantified. Using a designed sample of satellite-based observations of historical land cover change, we estimate the areas and trends in nine land cover classes from 1998 to 2017 in Ethiopia, Kenya, Uganda, Malawi, Rwanda, Tanzania, and Zambia. Our analysis found an 18,154,000 (±1,580,000) ha, or 34.8%, increase in the area of cropland in East Africa. Conversion occurred primarily from Open Grasslands, Wooded Grasslands, and Open Forests, causing a large-scale reduction in woody vegetation classes. We observed far more conversion (by approximately 20 million hectares) of woody classes to less-woody classes than succession in the direction of increasing trees and shrubs. Spatial patterns within our sample highlight regional land cover conversion hotspots, such as the Central Zambezian Miombo Woodlands, as potential areas of concern related to the conservation of natural ecosystems. Our findings reflect a rapidly growing population that is moving into new areas, with a 43.5% increase in the area of Settlements over the three-decade period. Our results show the areas and ecoregions most impacted by three decades of human development, both spatially and statistically.

**Keywords:** land cover change; TimeSync; East Africa; Landsat; statistical inference; development

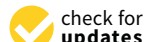



## 1. Introduction

East Africa has undergone rapid economic growth and environmental change over the past 30 years, resulting in considerable gains in livelihood, sometimes at the expense of natural ecosystems. From 1990 to 2020, average life expectancy increased from 45 to 67 years, while forest cover decreased by 17% [1,2]. Meanwhile, climate change is changing human-environment interactions in complex and often poorly understood ways. For example, crop productivity and food security issues arise under a changing climate, while woody encroachment may be accelerated in part by carbon fertilization [3–6].

Land cover change in Sub-Saharan and interior East Africa (referred to here as "East Africa" for brevity and consisting of the nations of Kenya, Malawi, Rwanda, Tanzania, Ethiopia, Burundi, Zambia, and Uganda) often involves the conversion of woody natural habitats to less-woody cultivated or developed land cover types. The region contains 60% of the global uncultivated arable land and a population expected to double by 2050 [7,8]. Consequently, land use intensification will continue to support the growing population, affecting some of the most environmentally important and threatened ecosystems in the world. Approximately 73% of the region is contained in the World Wildlife Fund's (WWF) list of 200 ecoregions prioritized for conservation, compared to 31% of the continent as a whole [9].

Land cover in the region is composed primarily of tropical and subtropical grasslands, shrublands, and savannas. Given that grassland, shrublands, and savannas store only approximately 40% of the carbon sequestered by tropical forests per unit area, these areas have not been the focus of conservation mechanisms aimed at offsetting human carbon emissions [10]. However, their soils are also much less productive than in tropical forests, therefore requiring more area of land conversion to achieve similar agricultural yields [11]. Despite having fewer trees than forests, the region's savannas still contain approximately 5.5 PgC in aboveground carbon [12]. This carbon storage is at risk as pressures toward agricultural and urban expansion persist.

However, there is evidence that additional carbon sequestration is occurring in the region through woody expansion in grasslands, caused in part by increasing atmospheric $CO_2$, fires, changing hydrologic conditions, and shifting ecozones [4,13]. Using L-band radar analysis, McNicol et al. (2018) found that between 2007 and 2010, carbon gain in Sub-Saharan woodlands largely counteracted loss, due to degradation and deforestation. Brandt et al. (2017) used passive remote sensing to reach a similar conclusion, finding an increase in woody cover between 1992 and 2011 in Sub-Saharan Africa, largely in drylands [14]. Objective monitoring of trends among the Region's land cover classes (which here include Cropland, Dense and Open Forests, Vegetated Wetland, Wooded and Open Grasslands, Settlements, Open Water, and Other) is needed to reconcile these results with rapid population growth and dependence on fuelwood for the Region's energy [15].

Remote sensing represents the most feasible way of obtaining land cover information in a temporally and spatially consistent way. To date, most studies on land use and cover change in the region have been limited to coarse resolution analysis (e.g., Reference [16]), smaller spatial domains (e.g., References [17–19]), or have involved only change affecting a particular cover type, such as forests [20,21]. The limited spatial domains and/or change classes analyzed in these studies prevent the comprehensive and regional assessment of trends in land cover change. Here, we seek to fill this information gap by analyzing long-term (30 years) land cover trends using a legend, spatial domain, and time period that reflect the region's ecological diversity and gradual socioeconomic and climatic changes.

A tool that has proven appropriate for coherent land change research is TimeSync, which provides users with a platform for interpreting historical Landsat data to record land change dynamics at a designed statistical sample of reference locations [22,23]. International "Good Practice" guidance for monitoring land change recommends using such a sample with an unbiased statistical estimator [24–26]. This allows consistent estimates of the land cover distribution of an area of interest over time and with straightforward measures of uncertainty.

Area estimation through formal inference using a reference sample has proven advantages beyond simple map-based approaches to estimating area (i.e., "pixel counting"). The primary disadvantage to "pixel counting" is that map errors can introduce a substantial bias to the estimates of area, which is particularly true when evaluating change in complex landscapes like in East Africa [24–26]. Implementing a designed sample of high-quality observations allows for using unbiased statistical estimators and straightforward uncertainty estimation. While this approach does not generate a high-resolution map of land cover change, changes observed across a sampling grid can nevertheless be used with kernel-based "heat map" approaches to understanding sub-regional spatial patterns of change.

This research is aimed to address the following research question: What are the primary land cover and land use trends in East Africa over the previous 30 years, and how do they vary by country and ecoregion? To address this question, we performed a robust regional assessment using TimeSync as implemented by regional land cover experts.

## 2. Methodology

### 2.1. Study Area

We analyzed trends from 1988 to 2017 in the East African nations of Uganda, Rwanda, Tanzania, Malawi, Zambia, Kenya, and Ethiopia (Figure 1). Predominant ecoregions in the study area include the miombo woodlands in Zambia, Tanzania, and Malawi, dense tropical forests in Rwanda and Uganda, montane shrublands in Ethiopia, and subtropical savannas. Apart from Rwanda in the 1990s, each country has undergone relatively consistent economic and population growth (World Bank, 2014). To support this growing population and to combat a recent legacy of food insecurity, the region is actively expanding and intensifying agricultural activity. The three-decade study period was chosen to correspond to the maximum time of socioeconomic growth with consistent reference data necessary for the analysis (described below).

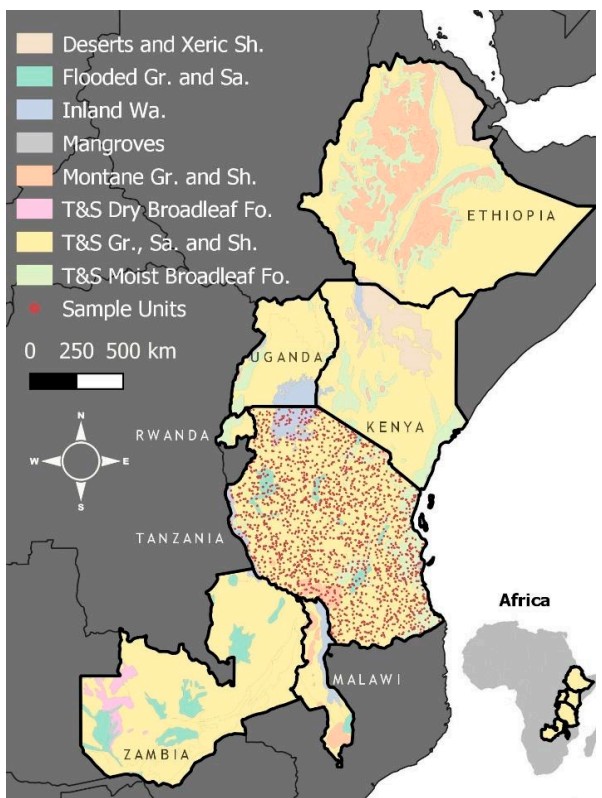

**Figure 1.** The ecoregions of the seven-country study region in East Africa. The 2000 randomly selected sample units are shown for Tanzania, however the same sampling routine was performed in each country. Sh, Shrublands; Gr, Grasslands; Sa, Savannas; Fo, Forests; T&S, Tropical and Subtropical. Land covers are defined by ecoregion categorization in Olson et al. (2001).

### 2.2. Area Estimation

Land cover areas were estimated using observations of land cover history at a simple random sample of 2000 locations in each country. The spatial and spectral resolution of Landsat imagery, in addition to its consistent availability back in time, make it ideal for this application. Sample points were interpreted by local experts using the TimeSync software [22], accessing historical Landsat and high-resolution imagery. A land cover label was assigned to each sample unit for every year from 1988 to 2017 by evaluating the sample location against all available Landsat data and high-resolution imagery on Google Earth. The reference labels represent sub-classes to the International Panel on Climate Change (IPCC) top-level land use categories: Open Forest, Dense Forest, Cropland, Settlements, Wooded Grassland, Open Grassland, Vegetated Wetland, Open Water, and Other land (definitions given in Reference [27], and elaborated in Table 1).

**Table 1.** Land cover class definitions are derived as sub-classes from the International Panel on Climate Change (IPCC) top-level land use categories.

| Land Cover Class | Definition |
| --- | --- |
| Open Forest | Tree-covered areas with 15–40% canopy cover. This class includes both natural and planted forests that meet the canopy cover threshold and the national definitions of forest cover. |
| Dense Forest | Tree-covered areas with over 40% canopy cover. This class includes both natural and planted forests that meet the canopy cover threshold. |
| Cropland | An area primarily used for agriculture. Includes field crops, agroforestry, horticulture, perennial crops and floriculture, and abandoned croplands left fallow. |
| Settlements | Areas primarily built for the development of buildings, roads, or other human-modified land uses. |
| Wooded Grassland | A natural landscape that does not meet the tree canopy thresholds of either forest class and contains woody vegetation in the form of shrubs, very sparse tree cover, or single trees. |
| Open Grassland | A natural landscape that does not meet the tree canopy thresholds of either forest class and does not containing substantial woody vegetation. The plant cover is composed principally of grasses, grass-like plants, and forbs, including areas where practices such as clearing, burning, chaining, and/or chemicals are applied to maintain the grass vegetation. |
| Vegetated Wetland | Areas covered in water for at least part of the year and containing a mix of vegetation and open water. Includes marshlands, swamps, and peatlands. |
| Open Water | Open water bodies, including oceans, lakes, rivers, and reservoirs. |
| Other land | Areas that do not meet a different class definition, including barren land, rock outcrops, permanent snow cover, beaches, and salt crusts. |

The reference samples informed our analysis in two ways. First, the samples were used as the basis for clustering using weighted kernel density analysis. Clustering was performed in Python using the 'KernelDensity' estimator (parameters: bandwidth = *0.2*, kernel = *gaussian,* algorithm = *ball_tree*, additional parameters = *default*) in the Scikit-learn module [28]. The purpose of this analysis was to determine and visualize "hotspots" of important land use dynamics. To account for differences in country sizes, each sample unit was given a weight based on the proportional size of the country containing the sample. For each sample unit *n* in country *c,* the weight was calculated as:

$$w_{c,n} = 1 * \left( \frac{a_c}{a_{max}} \right)$$

where,

$$w_{c,n} = sample\ weight\ for\ sample\ unit\ n\ and\ country\ c$$

$$a_c = Area\ of\ country\ c$$

$$a_{max} = Maximum\ area\ of\ countries\ in\ study\ region$$

The second use of the reference samples was to statistically estimate areas of land cover and change at various spatial scales using statistical inference. Areas were estimated at the country, regional, and ecoregion scale. For each scale of analysis, it was necessary to properly account for the inclusion probability of each sample unit for the corresponding spatial domain. The samples were selected at the country scale. Therefore, the inclusion probability of each sample unit is directly dependent on the area of the country. For estimation at the country scale, each sample unit has the same inclusion probability, and the area of classes of interest can be estimated directly based on the proportion of each class within the reference sample [29]. In this context, areas and standard errors can be calculated using a simple expansion estimator, with the implementation used here defined in Equations (29) and (30) in Reference [30].

To estimate areas across the entire seven-country study region, the expansion estimator could not be utilized, since the inclusion probability differs for sample units of different countries. To account for varying inclusion probabilities, we deployed a stratified estimator

(Equations (31) and (32) in Reference [30]) for the regional analysis in which the area weights of each sample unit were computed from the size of each country. For estimation within ecoregions, which cross national boundaries and contain differing sample subsets, the ratio estimator described in Stehman (2014) was used [31].

### 3. Results

Our analysis revealed 29,040,000 (±1,950,000) ha of land cover change from 1988 to 2017, representing 7.7% of the study region (Figure 2). Proportionally, the areas of Cropland and Settlement increased by 35% and 43%, respectively, while all other land use classes decreased (Figure 3). In 2017, the dominant land cover in the region was Wooded Grassland, which, despite a 7.0% decrease over the study period, still made up 36% of the study area, or 134,400,000 (±3,710,000) ha. Altogether, there was a decline of 18,940,000 (±1,600,000) ha naturally vegetated land uses (Grasslands, Forests, and Vegetated Wetland).

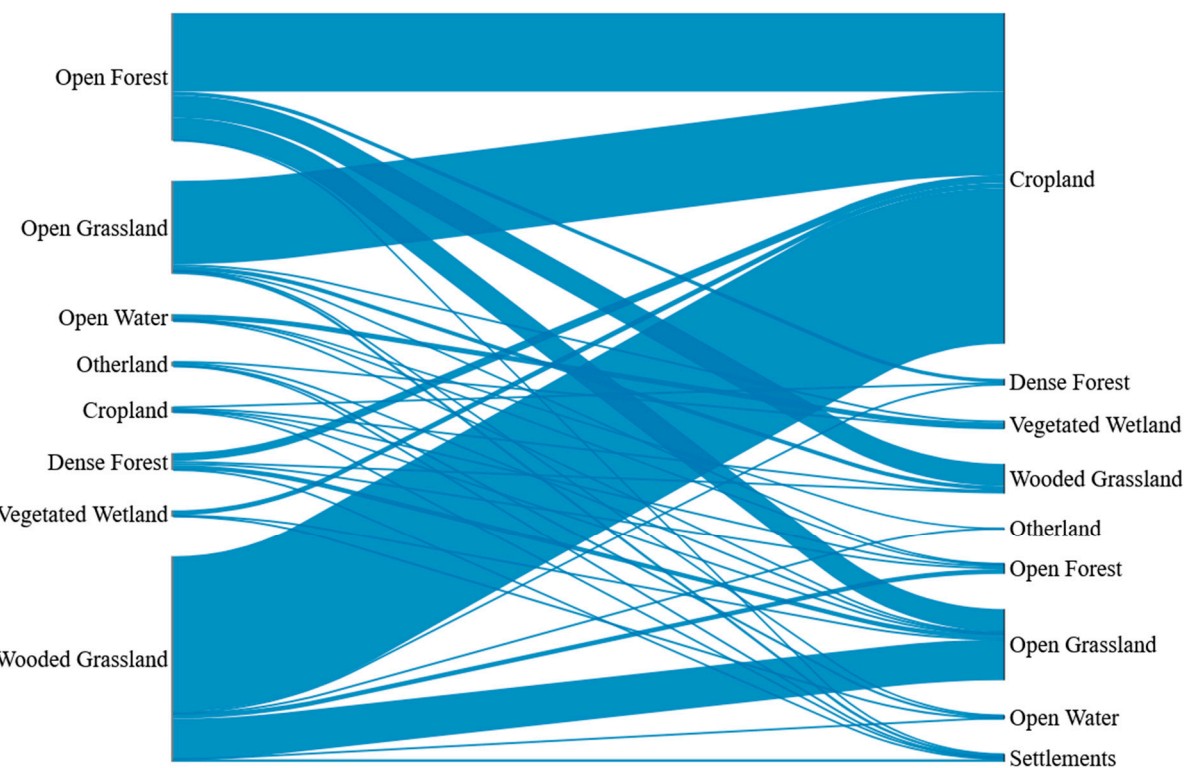

**Figure 2.** A Sankey diagram, showing land use transitions between 1988 (left) and 2017 (right). Total change class area (e.g., loss of Open Forest) corresponds to the size of the class nodes, while the link width corresponds to the area of conversion between classes. Note that the largest links (or greatest area of conversion) exist between the conversion of Wooded Grassland, Open Grass-land, and Open Forest to Cropland.

Land cover change as a proportion of land area was greatest in Uganda (5.29% ± 1.54) and Malawi (4.21% ± 1.56), and lowest in Kenya (0.86% ± 0.47) and Ethiopia (2.49% ± 1.38) (Figure 4). Land cover conversion in Uganda was greatly accelerated after 2000, after which there was a notable increase in Cropland and reduction in Open and Wooded Grasslands. Over the 30-year study period, the most common land cover in Uganda switched from Open Grassland to Cropland, while in Zambia, it switched from Open Forest to Open Grassland. No other country changed the primary land cover composition. The land covers with significant changes in the area are detailed below.

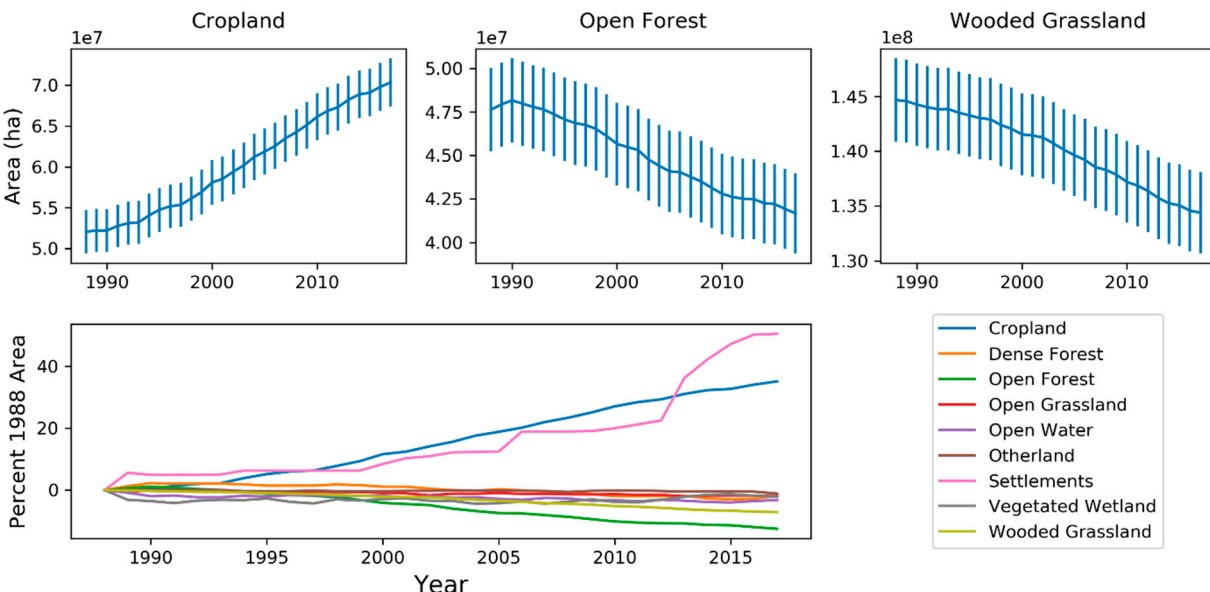

**Figure 3.** Yearly change in the three classes with the greatest overall change (top), and proportional change in land cover classes across the study domain relative to the area in 1988 (bottom). Standard errors are expressed as vertical lines in the plots of yearly estimated areas.

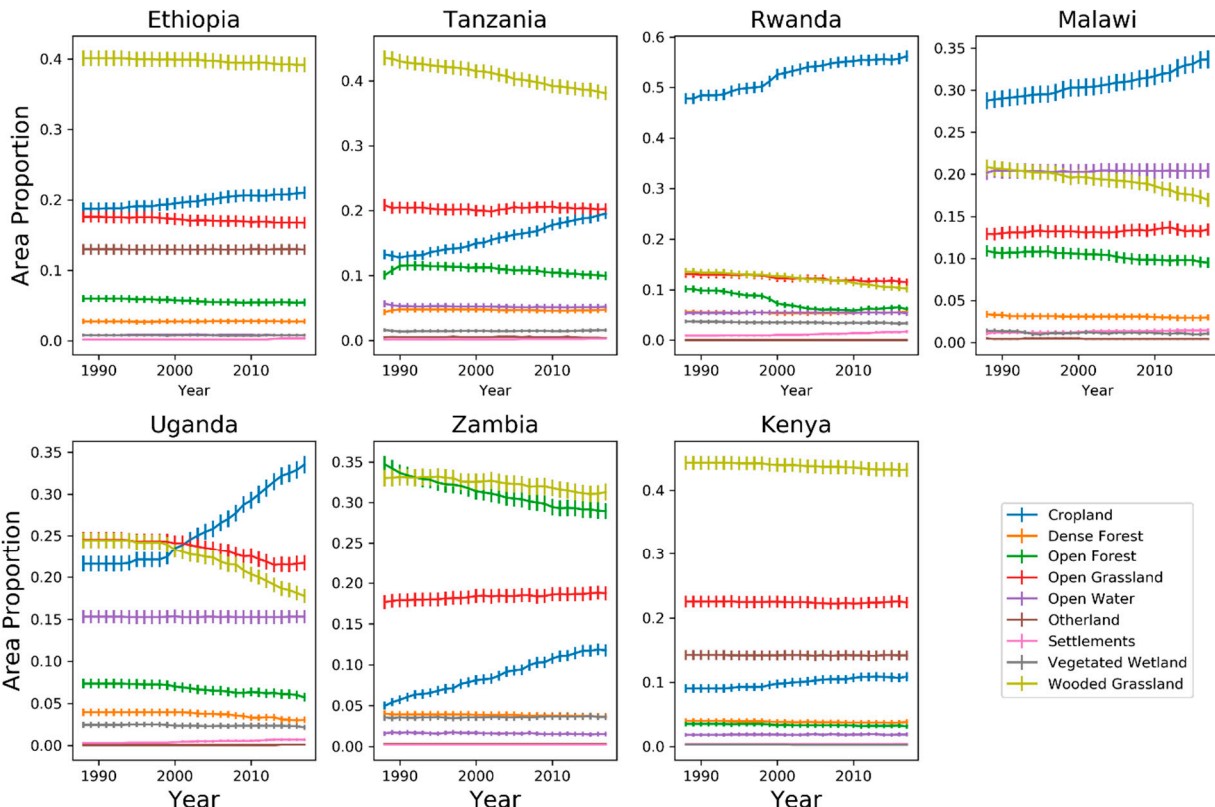

**Figure 4.** The yearly proportion of land cover classes for each country in the study region. Note that Uganda and Zambia changed in the land cover representing the largest proportion of the country.

### 3.1. Conversion to Croplands

In total, there was 18,154,000 (±1,580,000) ha, or 34.8%, increase in Cropland. According to our sample, the development of Cropland was most prevalent in Open Grasslands, followed by Wooded Grassland and Open Forests. Primary hotspots for the conversion of Cropland in addition to the largest proportional increases occurred in Zambia (106%), Uganda (55%), and Tanzania (53%), while the largest overall area increase was in Tanzania (64,10,000 ha). In the last decade of our analysis, the fastest growth in Cropland occurred in Uganda (0.4%/year) and Zambia (0.27%/year). The highest agricultural conversion rate occurred in the Central Zambezian Miombo Woodlands (±5,940,000 ha), Victoria Basin Forest-Savanna Mosaic (±1,980,000 ha), Southern Acacia-Commiphora Bushlands and Thickets (±1,600,000 ha), East Sudanian Savanna (±1,300,000 ha), and the Eastern Miombo Woodlands (±1,100,000 ha) (Figure 5).

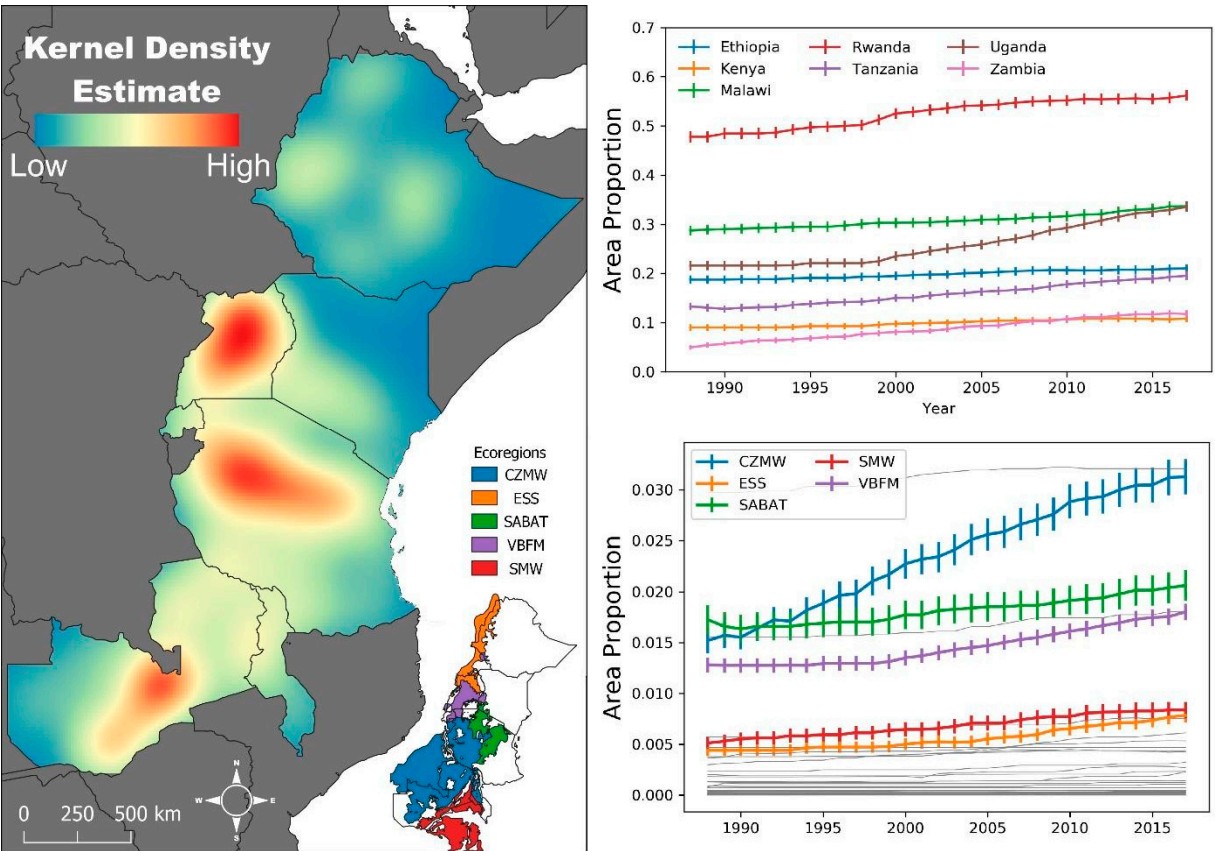

**Figure 5.** Cropland gain in East Africa. Left: Kernel Density hotspot analysis of new cropland. Top, proportional change of Cropland in seven countries. Bottom, Proportional change of Cropland in ecoregions in which the five ecoregions with the greatest change are colored (CZMW, Central Zambezian Miombo Woodlands, ESS, East Sudanian Savanna, SMW, Southern Miombo Woodlands, SABAT, Southern Acacia-Commiphora Bushlands and Thicklets, VBFM, Victoria Basin Forest-Savanna Mosaic). The grey lines represent the other ecoregions.

### 3.2. Settlements Expansion

The area of Settlements grew by 460,000 (±250,000) ha, or 43.5%, over the study period. Settlements increased in every country, but increased the most in Uganda, Rwanda, and Ethiopia, with smaller hotspots in Tanzania and southern Malawi (Figure 6). The primary land use replaced by Settlements for the region was Open Grassland, however the patterns were different in individual countries (Figure 7). In Rwanda, Settlements were developed in Wooded Grasslands and Cropland, while in Kenya, they were the result of deforestation of Dense Forest.

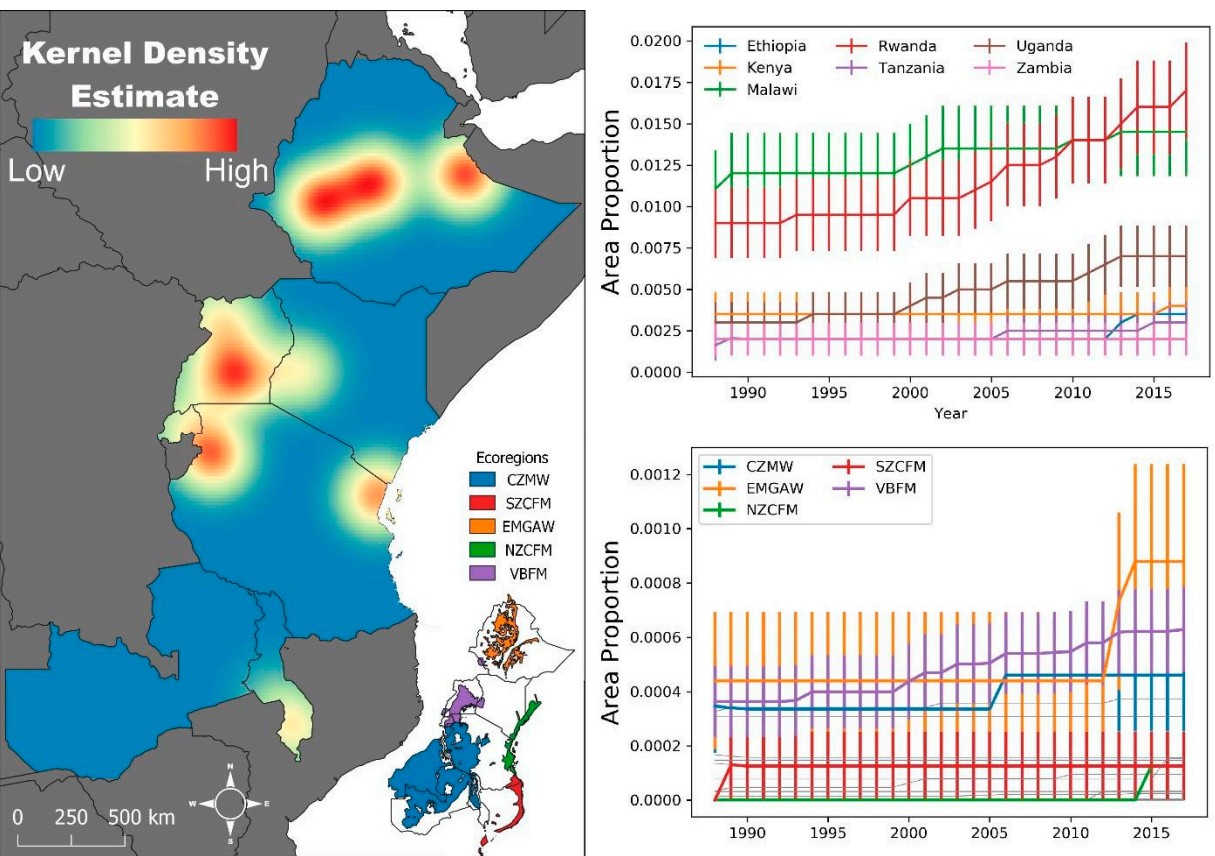

**Figure 6.** Settlement gain in East Africa. Left: Kernel Density hotspot analysis of new Settlements. Top, proportional change of Settlements in seven countries. Bottom, Proportional change of Settlements in ecoregions in which the five ecoregions with the greatest change are colored (CZMW, Central Zambezian Miombo Woodlands, VBFM, Victoria Basin Forest-Savanna Mosaic, SZCFM, Southern Zanzibar-Inhambane Coastal Forest Mosaic, NZCFM, Northern Zanzibar-Inhambane coastal forest mosaic, EMGAW, Ethiopian montane grasslands and woodlands). The grey lines represent the other ecoregions.

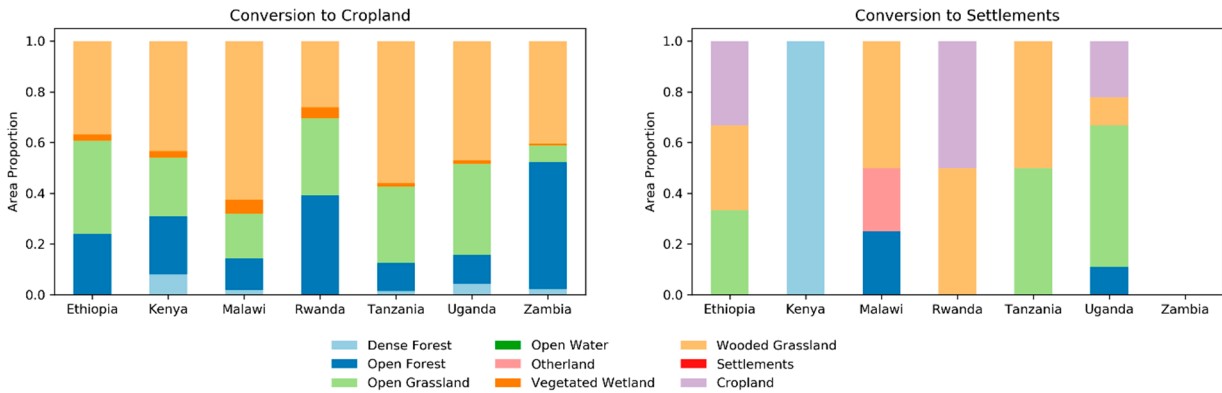

**Figure 7.** Previous cover type of land converted to Cropland (**left**) and Settlements, as a proportion of observed conversions per country.

### 3.3. Deforestation

Deforestation of Open Forest occurred most frequently in Zambia and Tanzania and resulted in a 13.5%, or 7,940,000 (±1,070,000) ha, decrease in Open Forests. The ecoregion with the greatest decrease in Open Forests was the Central Zambezian Miombo Woodlands (Figure 8). Dense Forests are much less common than Open Forests in the study domain, and therefore, deforestation in Dense Forests was also much less frequent, decreasing

950,000 (±330,000) ha. Deforestation in Dense Forests occurred most frequently in Uganda and was driven largely by conversion to Open and Closed Grasslands and Cropland (Figure 9). Despite substantial Dense Forest in Rwanda, no loss of such forests was recorded (1988–2017) in the 2000 study plots in that country.

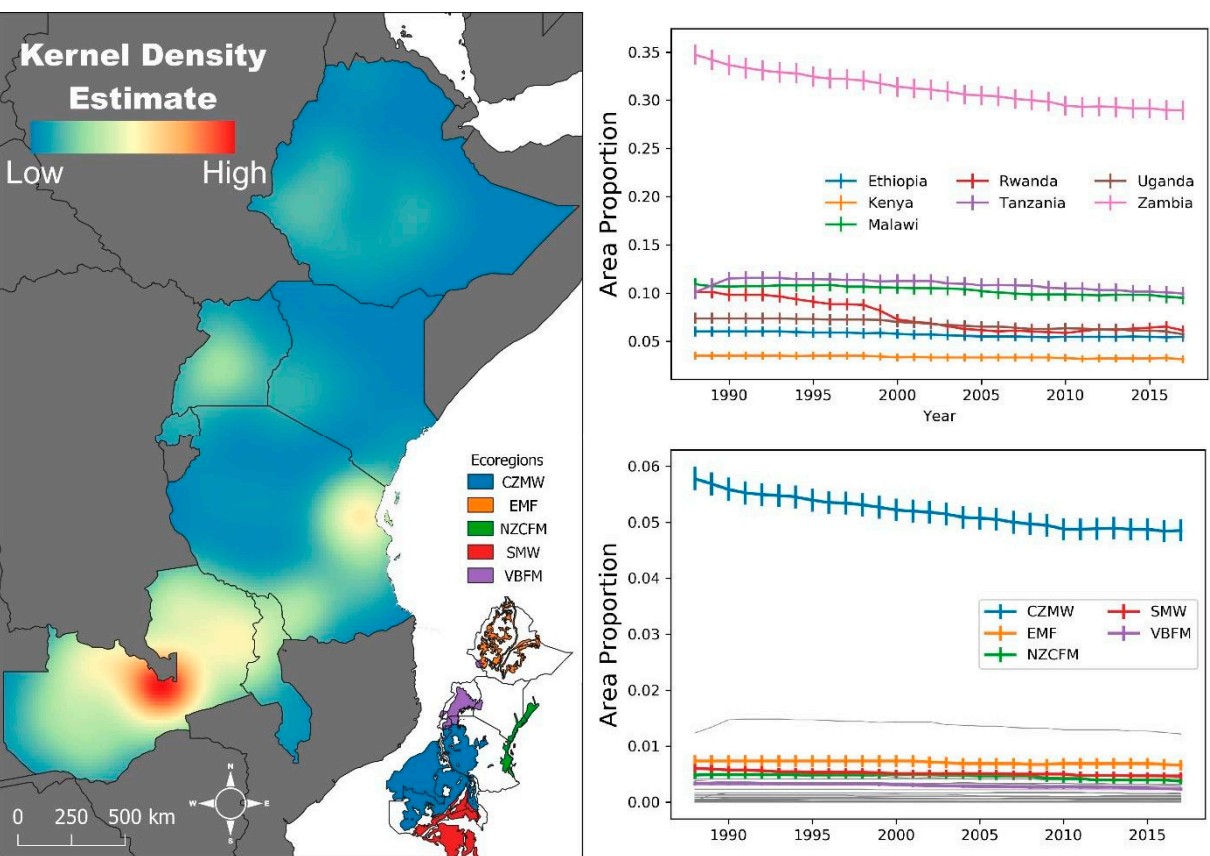

**Figure 8.** Deforestation in Open Forests in East Africa. Left: Kernel Density hotspot analysis of deforestation in Open Forests. Top, proportional change of Open Forest in seven countries with 95% confidence intervals expressed as vertical lines. Bottom, Proportional change of Open Forest in ecoregions in which the five ecoregions with the greatest change are colored (CZMW, Central Zambezian Miombo Woodlands, VBFM, Victoria Basin Forest-Savanna Mosaic, EMF, Ethiopian montane forests, SMW, Southern miombo woodlands, NZCFM, Northern Zanzibar-Inhambane coastal forest mosaic). The grey lines represent the other ecoregions.

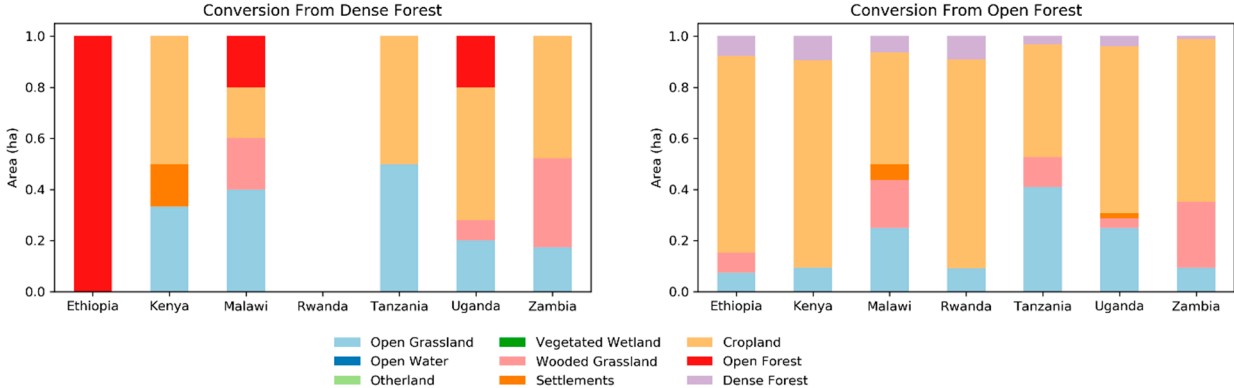

**Figure 9.** The proportion of land cover area after deforestation of Dense Forest (**left**) and Open Forest (**right**).

### 3.4. Wooded Grasslands

There were 12,170,000 (±1,330,000) ha of Wooded Grasslands converted to other land cover classes and 1,880,000 (±510,000) ha gain of Wooded Grasslands, resulting in a net change of 7.0%. Most (81%) of the conversion to Wooded Grasslands occurred in Open or Closed Forests, with the remaining occurring in Open Grasslands (14%) and Croplands (5%). The largest proportional decrease in ecoregions occurred in the Central Zambezian Miombo Woodlands, and by country was in Uganda (Figure 10). The only ecoregion with a net gain was the Zambezian Cryptosepalum dry forests; every other ecoregion was stable or decreasing in area. Conversion of Woody Grassland was predominantly for Cropland, and to a lesser extent, Open Grassland (Figure 2).

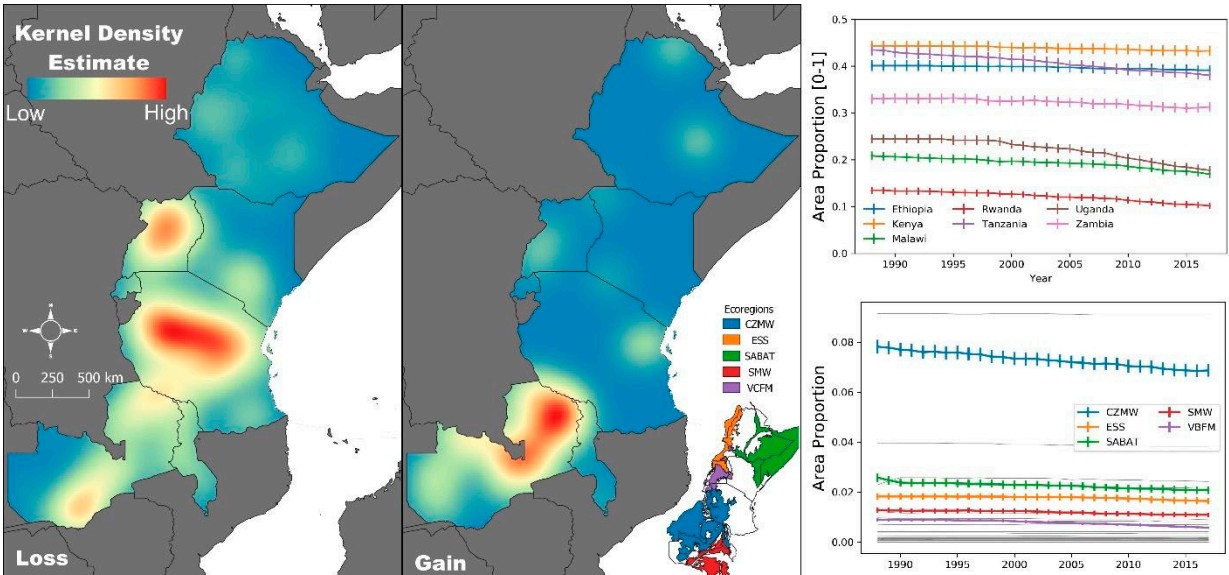

**Figure 10.** Heatmaps of loss and gain of Wooded Grassland. Left: Kernel Density hotpot analysis of loss and gain of Wooded Grasslands. Top: Proportional change of Wooded Grasslands in seven countries. Bottom: Proportional change of Wooded Grasslands in ecoregions in which the five ecoregions with the greatest change are colored (CZMW, Central Zambezian Miombo Woodlands, SMW, Southern miombo woodlands, ESS, East Sudanian savanna, VBFM, Victorian Basin forest-savanna mosaic, SABAT, Southern Acacia-Commiphora Bushlands and Thicklets). The grey lines represent the other ecoregions.

### 3.5. Cross-Class Woody Vegetation Change

The natural vegetation cover classes considered here contain a continuum of woody plant density. Specifically, classes in descending order of woody vegetation are: Dense Forest > Open Forest > Wooded Grassland > All other classes. This succession provides useful context for our estimates of cover class change. In isolation, an increase in Open Forests might be considered a positive development unless most new Open Forests come from the degradation of Dense Forest (which is the case, regionally). If we define woody vegetation change across classes according to the above order, our sample suggests that such change has been dominated by changes resulting in fewer trees and shrubs. Specifically, we estimate that changes involving reduced woody vegetation affected 20,930,000 (± 2,090,000) ha in the region, while changes resulting in more wood occurred on 940,000 (± 360,000) ha. Out of the 6,610,000 (± 980,000) ha change in natural vegetation classes (i.e., between non-developed classes, including grasslands, wetlands, and forests), 5,770,000 (± 1,150,000) resulted in a decrease in woody vegetation (Figure 11). Nowhere did the area of woody increase exceed woody decrease between natural or human-modified classes.

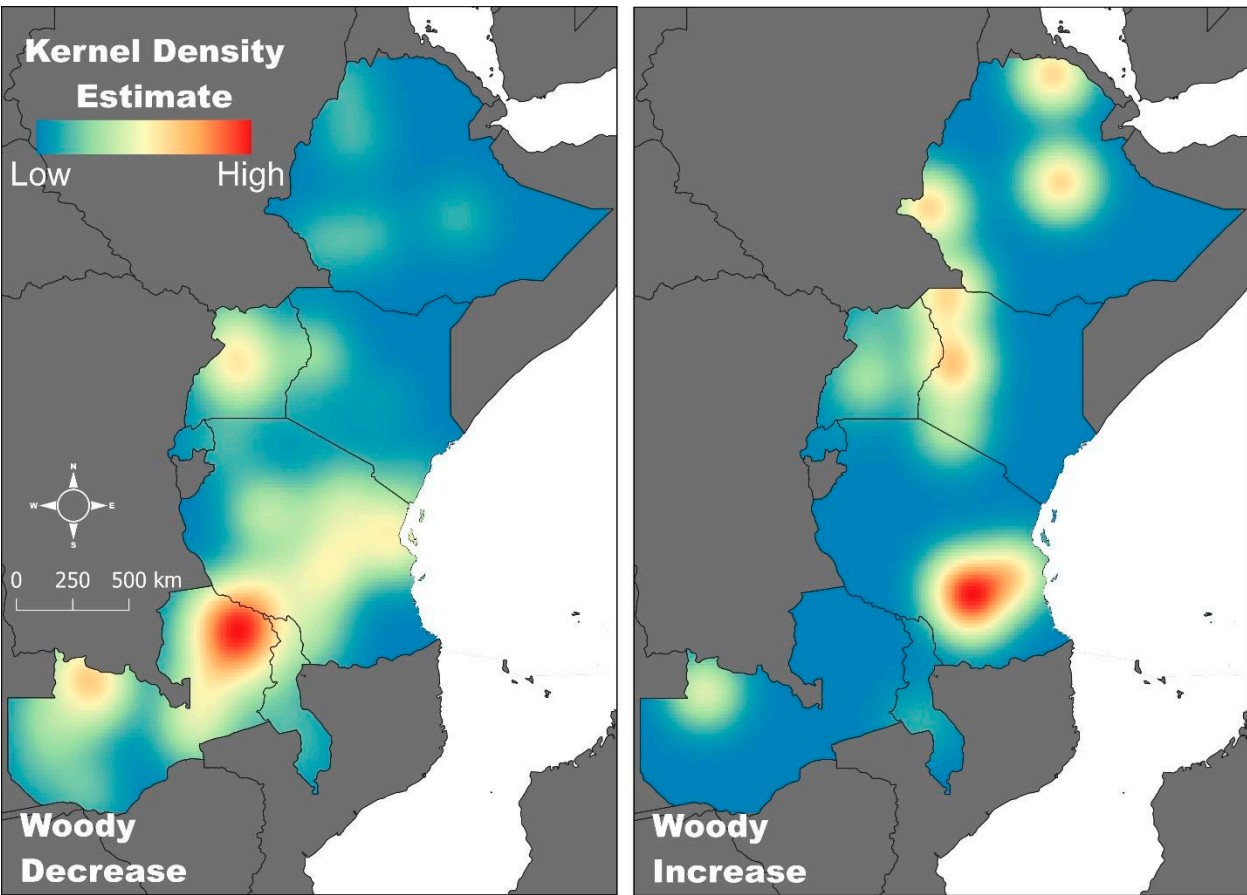

**Figure 11.** Hotspot analysis of land cover conversion resulting in a reduction in woody vegetation class (**left**) and increasing (**right**).

## 4. Discussion

East Africa has undergone extensive environmental change in the past three decades, largely driven by the expansion of Cropland and the conversion of naturally vegetated land covers. From 1988 to 2017, we found that the area of Cropland and Settlements increased by 35% and 43%, respectively, driving large-scale reductions in woody vegetation. While previously mentioned studies reported substantial increases in woody biomass sequestration in the region's ecosystems, land cover monitoring suggests a trend in the opposite direction.

A first-order approximation of an area's carbon dynamics that is commonly used in international reporting mechanisms (Penman et al., 2003) involves the calculation of emission factors associated with specific land cover conversions, which are then used with a record of such conversions (called "activity data") to infer changes in net carbon storage. It is outside the scope of this paper to assess whether within-class changes in the Region merit use of an evolving emission factor for the loss of natural woody cover, but the changes we document have clear carbon implications. McNicol et al. (2018) calculated that deforestation in woodlands in the Region resulted in a carbon decrease from $14 \pm 4$ to $6 \pm 5$ MgC ha$^{-1}$, with deforestation occurring most frequently in low-biomass parts of the landscape. Applying this estimate as an emission factor to our results would find approximately 0.015 PgC loss from conversion of Wooded Grasslands over our study period. While there were also gains in Wooded Grasslands, it was primarily at the expense of Open Forests, and therefore, would not represent substantial carbon gains. Future research can build on our results through more detailed carbon accounting to quantify landscape-level carbon dynamics.

All five of the ecoregions with the greatest increase in Cropland were savannas or woodlands. The ecoregion containing the largest proportion of land cover change was the Central Zambezian Miombo Woodlands. Due in part to its poor soils, the Zambezian Woodlands has remained largely intact and has consequently become a hotspot of biodiversity [11]. Of the global ecoregions, it contains the third-highest mammal species richness and 17th in floral diversity [32,33]. The Zambezian Woodlands are also located in the heart of the "savannah transformation frontier", with a rapidly growing population and increasing pressure from poaching, fuel harvesting, and agriculture [11]. This trend is emblematic of Sub-Saharan Africa as a whole: Growing demand for food is forcing agricultural expansion in historically less developed savannas and woodlands.

Every country in our study region saw the expansion of Settlements, with the largest proportional increase occurring in Rwanda. D'Amour et al. (2017) predicted that Rwanda, which has the highest population density of any continental African country, will lose 34% of its cropland, due to urbanization by 2030, more than any other country besides Egypt [34]. Currently, an estimated 31% of the nation's GDP originates from the agricultural sector, which suggests that economic and societal rearrangement would be necessary for large-scale urbanization in croplands [35]. Our results did reveal 10000 ha of Cropland converted to Settlements in Rwanda, proportionally the largest of any country (Figure 7). However, we found 10 times more area of new Cropland, resulting in a net gain of 97,000 ha. Any loss of agricultural land near settlements is still being more than offset by new cultivation of former grasslands and woodlands.

Open Forests had the greatest proportional decrease of any class in the study region. In Ethiopia, Tanzania, Malawi, Uganda, and Zambia, deforestation of Open Forests primarily resulted in conversion to Cropland and Open Grassland. We hypothesize that conversion to Open Forest is due to timber and fuelwood harvest, although further investigation is needed to support this theory. Alternative drivers of deforestation in the region besides agriculture and development include poaching, hunting, and fire management. Interestingly, no conversion of Dense Forestland was detected in our Rwandan sample. While Dense Forest cover loss has indeed occurred in the country, particularly during times of conflict [36], the fact that no such conversion was detected across 2000 randomly located plots suggests that remaining Dense Forests are restricted to effectively protected reserves.

In 2017, we estimate that 23.5% of the study region was anthropogenically modified in the form of Cropland or Settlements. Jacobson et al. (2015) found a similar study region to be 29.8% anthropogenically developed. The discrepancy can likely be attributed to differences in class definitions: Jacobson assigned any visual sign of anthropogenic land cover to that class, while we assigned labels based on dominant land cover. Our study adds important historical context, as we found 26.4% of the area of Cropland and Settlements in 2017 was converted from natural vegetation since 1988. In other words, over a quarter of anthropogenic land cover conversion in East Africa has occurred since 1988.

Our analysis revealed three primary hotspots of land cover change in the region: Central Zambia, north-central Tanzania, and central Uganda (Figure 12). This information can be used to guide conservation initiations and mitigate unsustainable land cover changes in the future. Targeted research on the underlying socioeconomic drivers of conversion in these hotspots is needed to enact policies that can reduce the impact of development-driven land change on the environment. The fact that these hotspots occur primarily in savannas and woodlands supports the theory of a "savannah transformation frontier" described in Estes et al. (2016).

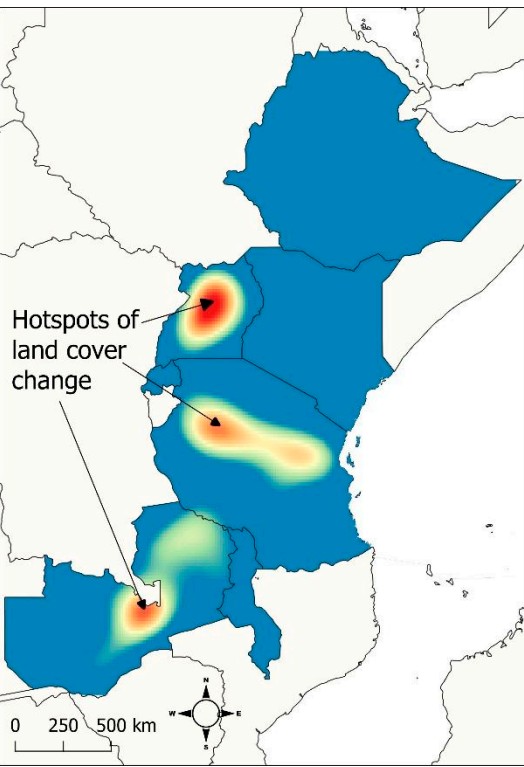

**Figure 12.** Three primary hotspots of land cover conversion, as calculated using kernel density estimation. This figure combines each type of land cover conversion analyzed in this study.

Our use of a designed sample of expert image interpretations calculates the confidence intervals for all our estimates in a straight-forward manner. While our estimates represent the most spatially and temporally consistent assessments of the region's land cover trends to date, our confidence intervals indicate how well our sample of 2000 random samples per country addresses specific classes of interest. For example, the margin of errors (defined as the ratio of the half-width of the 95% confidence intervals to the estimate and expressed here as a percentage) for our Settlement and Cropland change classes are 54.3% and 8.7%, respectively, indicating much higher precision in the estimate of Cropland change. This uncertainty framework is critical as local governments, aid agencies, and the private sector react to shifting land cover patterns by, for example, building new infrastructure and enacting natural resource protections.

Our findings show that land cover in the Region is already dynamic; substantial human-caused land cover change has already occurred in every ecoregion. The changes we document highlight new needs for infrastructure and suggest hotspots for future change. We also demonstrate that large shifts are occurring outside of forests. Greenhouse gas monitoring focusing only on deforestation will omit emissions from the conversion of systems like wooded grasslands. The uncertainty information provided by our statistical framework suggests the degree to which these observations apply at the level of particular domains, such as individual countries or ecosystems.

## 5. Conclusions

Effective land cover monitoring is necessary to address the environmental and social problems facing our planet. Here we used open access data and tools in conjunction with local environmental knowledge to monitor land use dynamics across thirty years in Eastern Africa. The results show clear patterns in land use intensification in the region: The development of cropland is expanding into previously intact savannas and woodlands, threatening critical ecosystems known for biodiversity and resulting in net reductions in woody vegetation.

**Author Contributions:** Conceptualization, S.P.H., Z.Y., and W.B.C.; methodology S.P.H., Z.Y., and W.B.C.; software, Z.Y., W.B.C., N.G., and E.L.B.; validation, E.L.B. and Z.Y.; formal analysis, E.L.B. and Z.Y.; investigation, P.O., N.G., Z.Y., S.O., E.O., W.B.C., S.P.H., and E.L.B.; resources, Z.Y. and N.G.; data curation, Z.Y. and E.L.B.; Writing—Original draft preparation, E.L.B.; Writing—Review and editing, S.P.H., Z.Y., and P.O.; visualization, E.L.B.; supervision, S.P.H.; project administration, S.P.H. funding acquisition, S.P.H.; All authors have read and agreed to the published version of the manuscript.

**Funding:** This research and APC were funded by NASA under SERVIR Applied Science Team grant NNH16AD021.

**Data Availability Statement:** The code and data used in this investigation are available on GitHub: https://github.com/bullocke/eastafrica.

**Acknowledgments:** We are grateful for the support provided by the SERVIR Science Coordination Office and the US Agency for International Development, and for the technical cooperation supported by Sylvia Wilson and the SilvaCarbon Program. The authors are also indebted to the RCMRD/SERVIR-Eastern and Southern Africa Project.

**Conflicts of Interest:** The authors declare no conflict of interest.

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
