# Peer review of "Three Decades of Land Cover Change in East Africa"

_land, doi:10.3390/land10020150_

Round 1

Reviewer 1 Report

This paper answers two questions: what are the primary land cover and land use trends in East Africa over the previous 30 years and how do they vary by country and ecoregion? Please make the below revisions:

  1. In the Introduction (Background) section, please talk more about the LULC classification methods (their advantages and disadvantages).
  2. For all the maps, please redo them. There are no scale bar, north arrow. And please pay attention to the legend location and background colors.

Reviewer 2 Report

The study aims to present the Research on “Three decades of land cover change in East Africa” The manuscript is presented clearly and nicely. I would like to suggest a Major revision for this paper. The paper needs significant modification before accepting for the journal. 

  1. Better to modify the topic.
  2. The abstract is very week. I need to provide a summary of the entire research. Same time primary purpose needs to be highlighted.
  3. It is better to discuss past attempts regarding the LULC in the selected counties. Why did you conduct the research? Why did you select 30 years? You have to explain it clearly.
  4. Figure 1 is not clear. You have to clearly show the essential elements of the maps such as scale, north arrow. It is better to show entire Africa as a small map in Figure 1. The names of the counties need to add.
  5. The classification of the LULC is not clear. Need more explanation
  6. All abbreviations need to be explained.
  7. The calculation of the conversion of cropland and settlements is not precise (Figure 7)
  8. Better to separate conclusion
  9. What is the accuracy of the findings? How this research helps to enhance the policy in the region.
  10. All maps need to modify by adding scale, legends, north, etc.

Reviewer 3 Report

The a good study on vegetation trends. We suggest changing the title Background to Introduction.

The Methodology is correct

Results. The trends of the different vegetation units as a consequence of the population increase in the countries studied have been well diagnosed.

Discussion and Conclusions. Discussion of conclusions should be separated, and since a good diagnosis has been made on the evolution of the vegetation cover, we suggest the authors to make concrete proposals in which economic-social development is contemplated in a sustainable way, and the different vegetation units.

Round 2

Reviewer 1 Report

I do not have further questions. Thank you.

Reviewer 2 Report

The paper has improved well. I think now the paper is good for publication. The authors had done excellent work to enhance the paper by considering reviewers' comments.